# Assessing the impact of a combined nutrition counselling and cash transfer intervention on women's empowerment in rural Bangladesh: a randomised control trial protocol

Elizabeth K Kirkwood ,[1] Michael John Dibley ,[1] John Frederick Hoddinott,[2] Tanvir Huda,[1] Tracey Lea Laba,[3] Tazeen Tahsina,[4] Mohammad Mehedi Hasan,[5] Afrin Iqbal,[5] Jasmin Khan,[5] Nazia Binte Ali,[5] Saad Ullah,[5] Nicholas Goodwin,[1] Sumithra Muthayya,[6] M Munirul Islam ,[7] Gulshan Ara,[7] Kingsley Emwinyore Agho,[8] Shams E Arifeen,[4] Ashraful Alam[1]

For numbered affiliations see end of article.

**Correspondence to**
Ms Elizabeth K Kirkwood;
elizabeth.kirkwood@sydney.edu.au

## ABSTRACT

**Introduction** There is growing interest in assessing the impact of health interventions, particularly when women are the focus of the intervention, on women's empowerment. Globally, research has shown that interventions targeting nutrition, health and economic development can affect women's empowerment. Evidence suggests that women's empowerment is also an underlying determinant of nutrition outcomes. Depending on the focus of the intervention, different domains of women's empowerment will be influenced, for example, an increase in nutritional knowledge, or greater control over income and access to resources.

**Objective** This study evaluates the impact of the Shonjibon Cash and Counselling (SCC) Trial that combines nutrition counselling and an unconditional cash transfer, delivered on a mobile platform, on women's empowerment in rural Bangladesh.

**Methods and analysis** We will use a mixed-methods approach, combining statistical analysis of quantitative data from 2840 women in a cluster randomised controlled trial examining the impact of nutrition behaviour change communications (BCCs) and cash transfers on child undernutrition. Pregnant participants will be given a smartphone with a customised app, delivering nutrition BCC messages, and will receive nutrition counselling via a call centre and an unconditional cash transfer. This study is a component of the SCC Trial and will measure women's empowerment using a composite indicator based on the Project-Level Women's Empowerment in Agriculture Index, with quantitative data collection at baseline and endline. Thematic analysis of qualitative data, collected through longitudinal interviews with women, husbands and mothers-in-law, will elicit a local understanding of women's empowerment and the linkages between the intervention and women's empowerment outcomes. This paper describes the study protocol to evaluate women's empowerment in a nutrition-specific and sensitive intervention using internationally validated, innovative tools and will help

## Strengths and limitations of this study

► This paper describes the study protocol evaluating women's empowerment in a nutrition-specific and nutrition-sensitive randomised controlled trial and will help fill the evidence gap on pathways of impact for women's empowerment and highlight areas to target for future policy and programming.

► Formative research guided this mixed-methods approach.

► We have designed specifically tailored tools, based on a theory of change, and using an internationally validated index that has been piloted in Bangladesh.

► We are not using the Project-Level Women's Empowerment in Agriculture Index in full as we interview women only for the quantitative component and will not calculate the Gender Parity Index, which is an index that sheds light on the sense of intra-household inequality; however, we will explore this with qualitative research.

fill the evidence gap on pathways of impact, highlighting areas to target for future programming.

**Ethics and dissemination** Ethical approval has been obtained from the International Centre for Diarrhoeal Disease Research (Ref: PR 17106) and The University of Sydney (Ref: 2019/840). Findings from this study will be shared in Bangladesh with dissemination sessions in-country and internationally at conferences, and will be published in peer-reviewed journals.

## BACKGROUND

To define women's empowerment is not straightforward, nor is it easy to measure across the different domains and phases of a woman's life. Empowerment is 'the process of enhancing an individual's or group's capacity to make purposive choices and to transform

BMJ

those choices into desired actions and outcomes'.[1] The United Nations defines women's empowerment as having five key components: 'women's sense of self-worth; their right to have and to determine choices; their right to have access to opportunities and resources; their right to have the power to control their own lives, both within and outside the home; and their ability to influence the direction of social change to create a more just social and economic order, nationally and internationally'.[2]

Gender, socially constructed roles assigned by sex, is one of the keynote determinants of health outcomes,[3] making gender equality and the rights of women an imperative goal in global public health. Gender inequality and restrictive gender norms are, according to the Lancet 2019 series on Gender Equality, Norms and Health, 'powerful but separate determinants of health and well-being'.[3] Sustainable Development Goal (SDG) 5 aims to achieve gender equality and to empower all women and girls, ending discrimination in all forms, from violence to forced marriage,and experience full active participation and equal rights in all spheres of life—politically, economically and socially.[4] Out of the 17 SDGs, 11 entail indicators related to gender.

Much progress has been made towards gender equality and the empowerment of women globally. Yet, the potential for women to fully participate as 'agents of change' is still limited due to persistent social, economic and political inequalities.[5] Women continue to be constrained by norms, beliefs and customs through which societies differentiate between women and men.[6] Contextual forms of oppression include patriarchal societies and institutions, as well as lack of opportunity due to socioeconomic circumstances such as poverty. Violence, including intimate partner violence (IPV), disproportionally affects women. Worldwide, a third of women report experience physical or sexual IPV.[7] Women subjected to IPV suffer poor mental and physical health, and their children show poorer health, nutrition and development outcomes.[8] Evidence suggests that the combination of cash transfers and behaviour change communication (BCC) can increase women's bargaining power and poverty-related emotional well-being and can lead to a reduction in IPV.[8]

Interventions to improve the health and nutrition literacy of women and, in particular, counselling through mobile phone applications can make a positive impact on women's empowerment. Mobile health (mHealth) interventions have the potential to improve a woman's confidence and decision-making skills in communicating with healthcare professionals. However, a systematic review revealed the need for a further rigorous investigation into mHealth, in terms of implementation and evaluation, to establish whether mHealth programmes transform rather than reinforce gender inequalities.[9] When a woman receives additional resources, such as cash transfers, and is the target of an mHealth programme, this can challenge gender norms within relationships and exacerbate gender disparities.[9] There is limited evidence on the adverse effects of mHealth, such as expanding the digital divide and gender-based power imbalances, and this lack of evidence emphasises the need to monitor the impact of mHealth interventions on gender relations and women's empowerment.[9]

The Lancet Series on Maternal and Child Nutrition also advocates the use of cash transfers and social safety net programmes, which are increasingly the preferred approach to support households living in chronic and extreme poverty.[10] When targeted at women, social protection can promote women's economic empowerment and enhance decision-making ability, with the overarching assumption that control over cash will lead to greater investment in children's health and education.[10 11] When women are engaged in nutrition-sensitive programmes that use social safety nets, certain aspects of women's empowerment are augmented, such as changes in gender roles and intrahousehold bargaining power.[10]

We plan to conduct a study, embedded in a cluster randomised control trial (RCT) that assesses a multifaceted nutrition intervention that aims to reduce childhood stunting—the Shonjibon Cash and Counselling (SCC) Trial.[12] This protocol paper presents the way we will measure the impact of the SCC Trial on women's empowerment.

## Developmental interventions and women's empowerment in Bangladesh

Although Bangladesh has made remarkable reductions in rates of poverty, almost one in four of the population still live in poverty (24.3% in 2016).[13] In 2018, the agriculture sector accounted for 40% of total employment.[14] In most low-income countries, women account for almost half of the agricultural labour force; however, in Bangladesh, women exceed 50% of the agricultural labour force.[15] In rural Bangladesh, social constructs such as gender roles of women and men powerfully drive household food consumption.[16] A nationally representative sample from Bangladesh found that women's empowerment increases dietary diversity and availability of calories at a household level.[17 18] The Government of Bangladesh and development organisations have shown a keen interest in combining nutrition counselling and cash transfers to improve maternal and child nutrition.[19]

In rural Bangladesh, 84% of households have access to a mobile phone.[20] A study in Bangladesh revealed women using Aponjon, a mobile messaging-based service providing information about mother and babies, had higher rates of autonomy in accessing mobile phones, although two-thirds of the women resided in urban areas. mHealth consultations may help address sociocultural gender norms and empower women as they were able to discuss personal health issues on the phone with female doctors who are infrequently found in rural communities.[20] The need to collect sex-aggregated, valid, comprehensive and standardised empowerment data is therefore imperative not only to achieve SDG 5 but also to enable the consistent measurement of women's empowerment indices.[21 22]

## Objectives and hypotheses

This study evaluates the impact of a combined nutrition counselling and an unconditional cash transfer RCT (SCC Trial), delivered on a mobile platform, on the level of empowerment of women in rural Bangladesh. This study will assess women's experience of empowerment and disempowerment throughout the SCC trial.

The primary hypothesis is that in a community-based cluster RCT of a mobile phone-based nutrition BCC and unconditional cash transfer given to women in low-income families in rural Bangladesh, women's empowerment will increase as measured by an increase in mean women's empowerment scores (by an average percentage of 20) from the baseline to the end of the 24-month intervention, compared with women in the control arm.

The secondary hypotheses are that the mobile phone nutrition BCC and unconditional cash transfer compared to usual programmes will (1) increase control over income and economic resources, (2) improve input and decision-making power in nutrition and health care choices, and (3) decrease the acceptance and experience of IPV.

## SCC Trial: women's empowerment theory of change

Empowering women is not an explicit goal of the SCC Trial; however, women gain access to nutrition counselling, cash transfers and a mobile phone; the use of these resources can empower women. We constructed the theory of change to illustrate how the SCC Trial will impact on women's empowerment outcomes. It outlines the barriers, intervention details, intermediate outcomes, outcomes and impact pathways (figure 1).

The SCC Trial intervention is delivered on a mobile platform. Women receive an mHealth interactive app—messages, audio and video, quizzes—as well as gestational age specific counselling from the call centre. They also receive cash via BKash mobile banking app. The principal rationale of the theory of change causal pathway to impact is that the SCC Trial will lead to an increase in women's empowerment for those women participating in the intervention arm.

The Project-Level Women's Empowerment in Agriculture Index (Pro-WEAI)[21] indicators, discussed in detail later, appear in the theory of change across the outcomes. We will use the six indicators from the Pro-WEAI to calculate the women's empowerment index and include (1) control over use of income, (2) input into productive decisions, (3) autonomy in income, (4) decision-making power on nutrition and healthcare, (5) respect among household members and (6) attitudes towards IPV.

One of the barriers to nutrition faced in rural Bangladesh is limited access to nutrition and infant and young child feeding (IYCF) information. Women in the SCC Trial intervention arm receive nutrition, IYCF and livelihood BCC delivered on a mobile app and nutrition counselling from the call centre. The intermediate outcome is woman's acquisition of new BCC/nutrition knowledge, contributing to increased input into and decision-making power about health and nutrition choices for herself and her family (indicator 4). Improvements in communication also occur (spousal, intrahousehold and community) as new knowledge is discussed and shared, including ways to put the messages into action (indicator 5). One's ability to make decisions is a measure of empowerment.[23] New knowledge can be used as leverage and bargaining power and enhance decision-making abilities across a range of areas such as livelihoods, household expenditure and decisions on productive assets, that is, what crops to farm (indicators 2–4).

The SCC Trial setting is in one of the poorest regions in Bangladesh. Families face barriers such as food insecurity and lack of financial resources. When a woman is the recipient of a cash transfer, the intermediate outcome is increased control over the use of income (indicator 1). Cash transfers, targeted at women, have the potential to improve livelihoods, alleviate poverty, enhance food security and increase women's empowerment.[10] Spousal communication is also strengthened as discussion ensues as to how to spend the

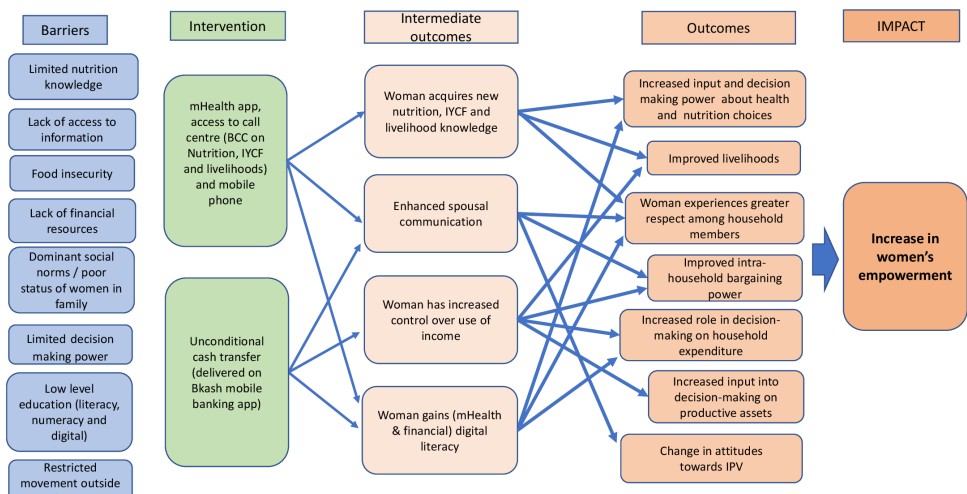

**Figure 1** SCC Trial: women's empowerment theory of change (adapted from De Silva et al[50]). BCC, behaviour change communication; IYCF, infant and young child feeding; SCC, Shonjibon Cash and Counselling.

additional income (indicator 4). The outcomes for women receiving an unconditional cash transfer include a strengthened ability to contribute to and participate in financial decision making. This decision making covers health and nutrition choices, household expenditure, productive assets, livelihood choices (ie, homestead gardening, buying chickens to raise for eggs, meat and income; indicators 1–4).

IPV includes behaviour that is physical, psychological, sexual, or abusive or controlling in nature.[24] There is evidence that the combination of additional income from cash transfers combined with BCC can increase women's bargaining power and poverty-related emotional well-being.[8] Changing attitudes, from within the family unit, as women participate and learn from intervention activities, has the potential to lead to a change in the acceptability of IPV, linked to a woman's right to bodily integrity.[25] Women may also experience an increase in respect and self-efficacy and subsequent reduction in acceptance of IPV (indicator 6).

## Study design
We will embed this study on women's empowerment within the SCC Trial. The SCC Trial will recruit pregnant women and follow-up over 24 months of an expected 2840 mother–child dyad, from recruitment in early pregnancy until the child is 18 months of age. The intervention will run for 36 months, depending on the time of enrolment and gestation, and the age of the child.

## METHODS
This study will use a mixed-methods approach with data gathered on women's empowerment from participants in the SCC Trial. This protocol paper will adhere to the SPIRIT (Standard Protocol Items: Recommendations for Interventional Trials) and Tidier guidelines for clinical trial protocols[26–28] to ensure rigour in protocol content.

## Study setting
We will conduct the study in two subdistricts of Sirajganj, Ullapara and Kamarkhanda, in northern Bangladesh (figure 2). Employment in Sirajganj is predominantly in agriculture (51 %), with 25% working in services and 23% in industry.[29]

## Eligibility criteria
All married women between 15 and 49 years, who are permanent residents of the study area and provide consent to participate and test positive with pregnancy urine test kit (Excel) (whose gestational age is ≤90 days), are eligible to participate in the SCC Trial.

## Intervention: Shonjibon[i] Cash and Counselling Trial
A separate protocol paper provides details of the design of the SCC Trial.[12] Briefly, the SCC Trial aims to assess the effectiveness of nutrition BCC (delivered via call centre counselling and specifically tailored mobile app) combined with unconditional cash transfers in reducing the prevalence of stunting (length for age<−2 Z) in children at 24 months. We will use a longitudinal cluster randomised controlled trial to determine the effectiveness of unconditional cash transfers and mobile BCC to reduce child undernutrition in Sirajganj district in northern Bangladesh, one of the poorest regions in the country according to the World Food Programme Poverty Maps.[30] Both quantitative and qualitative approaches will be employed to address the study objectives.

---

[i]Shonjibon stands for Shustho Notun Jibon in Bangla; translated, it means 'healthy new life'.

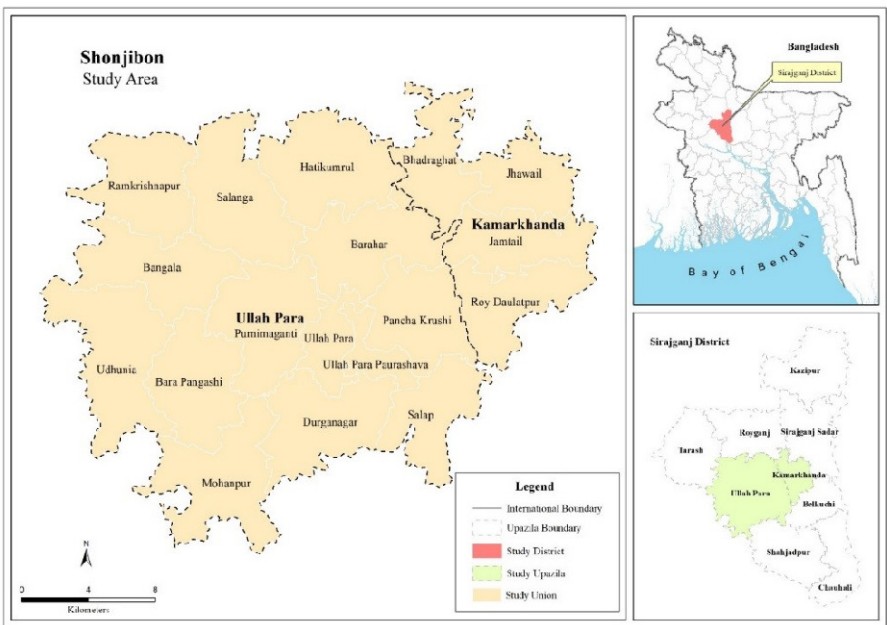

**Figure 2** Sirajganj, Bangladesh, Shonjibon Cash and Counselling study site.

The SCC Trial intervention arm receives (1) nutrition BCC delivered on a specially tailored app on a smartphone (audio, video and animation); (2) direct nutrition counselling from a call centre; and (3) unconditional cash transfer of 1000 Taka (US$12.50) received monthly via BKash mobile banking app. The control arm will receive a mobile phone and the current government of Bangladesh health and nutrition services. The participants in the control arm receive a mobile phone as the trial is interested in the effects of the messages, nutrition counselling and cash transfer delivered on a mobile platform, as opposed to the non-specific effects of a mobile phone.

## Study outcomes

The primary study outcome will be the difference in women's empowerment scores in the intervention arm as compared with the control arm of the SCC Trial. Secondary study outcomes will include (1) control over income and economic resources, (2) input and decision-making power in nutrition and healthcare choices, and (3) experience and attitude towards IPV.

## Outcome measurements

The SCC trial will collect data across the main module of the Pro-WEAI (12 indicators) as well as the additional health and Pro-WEAI Nutrition and Health Module. However, six of the Pro-WEAI indicators are likely to be directly impacted by the SCC trial; therefore, we will develop a composite women's empowerment indicator based on the following: (1) control over the use of income, (2) input in production decisions, (3) autonomy in income, (4) decision-making power about women and children's health and nutrition, (5) respect among household members and (6) attitudes towards IPV (online supplemental appendix table 1). We will also measure women's and family members' perceived and lived experiences related to women empowerment qualitatively (online supplemental appendix table 1).

1. Control over the use of income. We will measure control over the use of income by asking questions from Pro-WEAI; these questions are significant as the SCC Trial gives an unconditional cash transfer to the woman. Quantitative questions assess a woman's input into decision making on all sources of household income. Qualitative interviews will explore local perceptions of intrahousehold harmony and communication skills, issues related to withdrawing the cash, and perceptions and experience of control and expenditure of the money.
2. Input in productive decisions. These Pro-WEAI questions centre around input into household decision making on areas such as livelihood choices (activities the household participates in, such as agricultural activities, home gardens, etc), employment and household purchases.
3. Autonomy in income. This includes a series of vignettes, validated by cognitive testing, using a series of

hypothetical scenarios to assess motivation as to how income is used.
4. Decision-making ability about health and nutrition choices. We will examine responses from the Pro-WEAI Nutrition and Health Module regarding confidence and participation in decisions about nutrition and infant feeding, as well as a range of issues affecting women and children, such as whether to consult a doctor when ill to dietary choices when pregnant.
5. Respect among household members. Questions in this indicator pertain to a woman's spouse or other household members on respect and trust within relationships, and level of comfort in disagreeing with household members.
6. Intimate partner violence. Previous research in Bangladesh has found that cash transfers and BCC can reduce IPV.[8] We will measure a woman's attitude towards IPV using the Pro-WEAI as well as the direct experience of IPV using the questions from the Violence Against Woman and Girls 2015 Survey.[31 32] We will also conduct in-depth interviews to explore the woman's attitudes and perceptions of IPV qualitatively.

## Participant recruitment and timeline

A household census in the study area will record the names and contact details of all women who agree to participate, and we will enter their details into an electronic system. The system will generate for each participant with a unique ID number and quick response code. SCC surveillance workers will then conduct door-to-door visits each month to identify women that have missed two menstrual periods in a row. The woman will then undertake a pregnancy test (using a urine test kit (Excel). We will invite women who test positive for pregnancy to participate in the study with appropriate informed written consent. Based on experience with our pilot study, we anticipate 95% of the mothers will consent. Qualitative and quantitative data will be gathered at the start of the intervention and at endline (online supplemental appendix table 2).

## Allocation of clusters

Our study site has over 1100 villages, which would be adequate to cover the anticipated sample size. Each cluster encompasses two to four villages and consists of ~1000 households and ~4500 population. We will use a fixed randomisation scheme to assign treatments to eligible clusters, with a uniform allocation ratio of treatments, stratified by union. The sample size will consist of 104 clusters, total women (n=2840) allocated to two parallel groups (n=1420) with 20 mother–infant dyads per cluster. Each cluster consists of one to four adjacent villages with a minimum population of 600 households per cluster with 454 villages in our study area for clustering. A subsample of the households from a purposively selected subset of clusters will be chosen for qualitative data.

The intervention and control groups will receive similar maternal and child health and nutrition services

from the government, and the critical difference will be the nutrition BCCs with cash transfers on a mobile platform received by women in the intervention arm. The study design will control for potential observed and unobserved confounding factors as there will be an adequate number of clusters randomly allocated to the treatment groups. The geographical separation of clusters will limit the contamination of the intervention arm. We will use STATA V17 software to generate the random allocation sequence. We are unable to mask the treatment arm due to the nature of the intervention.

## Sample size and power

The sample size for our trial is fixed by the primary hypothesis in the main SCC Trial, which estimated a total sample of 2184 mother–infant pairs from 104 clusters.[12] We can find no reports of intracluster correlation coefficients (ICCs) for the women's empowerment indicators we plan to use. Therefore, we estimate that the fixed trial sample size will provide at least 80% power to detect a 20% increase in women's empowerment, assuming 30% of women are empowered and a high ICC of 0.2. Assuming a lower ICC of 0.05, we will have 90% power to detect a 13% increase in women's empowerment.

## Data collection methods

The SCC Trial will collect key household data and nutrition indices, while this study collects data on women's empowerment using the Pro-WEAI, whose focus is measuring agency for which there are limited, if any, standardised measures.[21] We will include the Pro-WEAI Nutrition and Health Module and supplementary questions regarding IPV. The qualitative component will be tailored to context-specific issues and guided by formative research.

## Quantitative data

We will collect the women's empowerment indicators in household surveys at the baseline and the endline of the 24 month study in the intervention and non-intervention arms. We will pretest all questions before the commencement of the study. We will use the validated short version of Marlowe-Crowne's social desirability scale,[33] which is based on a subset of 13 items from the original scale. We will calculate a social desirability score by adding up the number of socially desirable answers, out of the 13 questions. The potential range of the score will be from 0 to 13, and we will create three categories with a score of 0–4 graded as a low score, 5–9 as a medium score and 10–14 as a high score (online supplemental appendix table 3). The questionnaire assesses whether social desirability bias or the tendency of respondents to answer questions in a way viewed favourably by the research team has influenced the data collected. We will train the field researchers in detecting social desirability bias in qualitative data and limit bias by properly introducing the study to the respondents, building rapport and asking context-specific probing questions following the recent

framework described by Bergen and Labonté.[34] We will follow the process throughout the research.

## Qualitative data

The study uses qualitative analysis to elicit a local understanding of women's empowerment and the linkages between the SCC Trial and women's empowerment outcomes. We will generate qualitative data on women's and men's perceived and experienced change in empowerment through longitudinal qualitative semistructured in-depth interviews. We will also interview key family members as suggested by the women, for example, husband or mother-in-law.

We will explore the myriad of ways that women and men perceive and describe empowerment in their communities. A purposive sample will be interviewed approximately 2 months after the start of the intervention and at the endline of the SCC Trial. Due to the highly sensitive nature of the topics, a separate team of interviewers, from a different Upazila, will be trained to collect data on women's empowerment and IPV. We will train the team in line with the ethical and safety procedures as per the WHO's guidelines.[35] The training will cover measures such as changing topics and asking a decoy nutrition question if anyone comes into the interview space and having details of referral and support services for women affected by or experiencing IPV.

### Pro-WEAI

The Pro-WEAI measures the empowerment of women, agency and inclusion in the agriculture sector.[21] It is a standardised, survey-based internationally validated index developed to adapt to project-specific contexts and has been piloted in Bangladesh.[17 21] This study uses Kabeer's conceptual framework of empowerment that focuses on the ability of individuals to make choices, 'a process of change during which those who have been denied the ability to make choices acquire such an ability'.[36 37] Agency or the ability of the individual to make considered choices is at the centre of the framework and revolves around inter-related dimensions of resources, agency and achievements. This framework underpins the Pro-WEAI.[21] This study will collect data from women, across 12 indicators that are equally weighted (figure 3), with the addition of the Project-specific Nutrition and Health Module.

### Intimate partner violence

The Pro-WEAI includes questions on a woman's attitude to IPV. We will collect additional data about personal experience and behaviour-specific questions regarding IPV, as opposed to her perceptions thereof. We took these questions from the WHO Violence Against Women and Girls Survey (Bangladesh 2015).[32] We will also ask questions relating to cyberviolence.[38]

### Health and nutrition

The Pro-WEAI offers the option of adding project-specific modules, and for this study, we will add the Nutrition

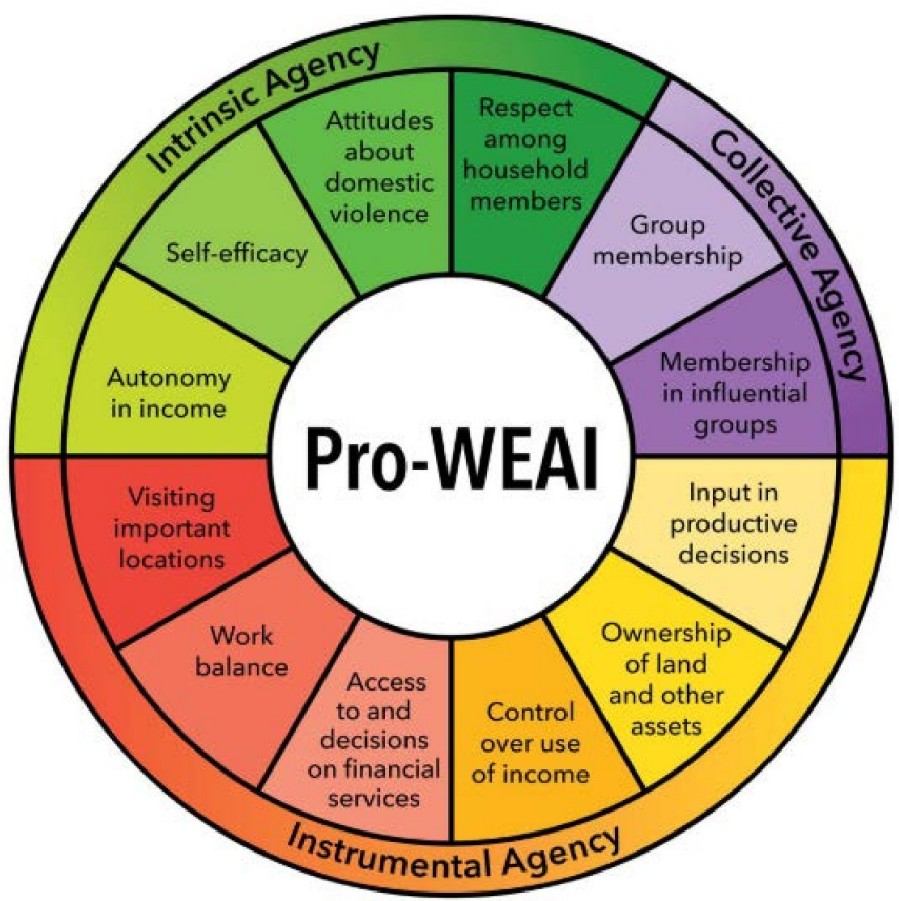

**Figure 3** Pro-WEAI: 3 domains of empowerment and 12 indicators.[21] WEAI, Project-Level Women's Empowerment in Agriculture Index.

and Health Module. This module asks questions about decision making about nutrition and health indices for women and children.

### Data collection and management

Intervention and evaluation staff will be separated from and will not know the trial hypothesis to minimise assessment bias. Android tablets will be used in the field to enable interactive data collection. During each evaluation interview, the data collectors will be guided and navigated by the software. As soon as the field staff complete each interview, they will upload the data to the server at International Centre for Diarrhoeal Disease Research (icddr, b). The data will be retained on the tablet and copied/merged with files at icddr, b, with android tabs being synchronised with the server daily. We will systematically scrutinise the captured data and address any discrepancies as they occur. Qualitative narrative data will be captured in digital audio recorders and stored in the server from where we will check for quality and improvement for future interviews.

### Quantitative data analysis

Data analysis will be by intention to treat. The women's empowerment composite index scores will be categorised as the percentage of women as empowered or disempowered. For each empowerment indicator, individuals are classified as adequate or inadequate based on Pro-WEAI predetermined thresholds. Women are considered empowered if four out of six indicators are adequate. We will also analyse impact of SCC intervention on individual indicators to assess increases or decreases in empowerment as not all indicators will respond or be impacted in the same way.

Analyses will be conducted at the mother–infant dyad level but will be adjusted for the cluster randomisation.[39] Primary analyses will compare the prevalence of women's empowerment at the end of the trial using Pearson's $\chi^2$ tests and 95% CIs for the group difference, adjusted for clustering and generalised linear mixed models for noncontinuous outcomes (eg, logistic mixed models for binary outcomes, for example, percentage of women's empowerment). Models will include treatment group as a fixed effect, infants as a random effect to account for repeated measurements, and community cluster as a random effect to account for cluster effects. We will also assess if the women's empowerment level at baseline, age, education, and presence or absence of a mother-in-law in the household modify the empowerment response to the intervention by testing for interactions between the intervention and these factors. STATA will be used for all analyses.

## Qualitative data analysis

We will record all interviews using digital audio recording devices. Interviews will be transcribed verbatim in Bangla by the research team at icddr, b. We will maintain quality control by checking the transcripts at random intervals to ensure the quality and accuracy of transcriptions. Data will be coded using thematic analysis approach.[40] We will use NVivo software to organise, code, categorise and compile the data. Qualitative methods offer an effective way of describing in-depth, complex and varied perceptions of empowerment within the local context. They will also aid with the interpretation of individual indicators within the quantitative data. Data triangulation by using different sources of data will add strength to the analysis.[41]

## Data and safety monitoring board

An independent data and safety monitoring board (DSMB) will be formed to assess the completeness and quality of data and to ensure data are compliant with recruitment and retention goals. The DSMB will also assess any factors that might affect the study outcome or compromise the confidentiality of the trial data. Any unintended effects of the trial will be reported to the board.

## Patient and public involvement

The development of the research question and outcome measures was informed by participants' priorities, experience and preferences based on our formative research and pilot study, whereby we engaged the local community and sought their views about the proposed intervention. Participants were involved in the design of the study by taking part in the formative research. Participants were not involved in the recruitment and conduct of the study. The key trial findings will be disseminated to study participants through meetings with community gatekeepers and local administration. In this RCT, the burden of the intervention was assessed by participants themselves as we undertook a pilot study.

## Dissemination plan

Lessons learnt from this study will be shared in Bangladesh with dissemination sessions in-country. Internationally, key findings will be shared with stakeholders, results will be presented at conferences and published in international peer-reviewed journals.

## Access to data

The SCC investigators will have access to all data and the right to analyse and publish data. Datasets will be shared after all personally identifiable information has been removed and to keep the identification of study subjects in confidence. The datasets generated and analysed during the study will be available from the corresponding author on reasonable request.

## Ethics approval and consent

We have obtained approval for the SCC Trial from the ethical review committee of the icddr,b (Ref. PR 17106), and the human ethics committees at The University of Sydney (Ref: 2019/840).

We will train the women on the use of smartphones as well as the nutrition counselling application. We will store the data obtained on the use of the application with access to limited authorised project staff. Trained field staff will educate the women about the safety procedures for receipt and use of money. We will record responses anonymously and by identification number. We will maintain the confidentiality, privacy and anonymity of the participants.

Project field workers will explain the nature, purpose, benefits, risks and process of the trial to potential participants as they register themselves in the surveillance system if they meet the inclusion criteria. We will train them to obtain written informed consent (in Bangla) from all intervention participants. Participation in the research will be voluntary, and the respondent will have every right to withdraw anytime throughout the programme without any obligation, loss or penalty. We will explain the participant's rights before collecting any data.

## DISCUSSION

The main goal of this study is to evaluate the impact of a combined nutrition counselling and unconditional cash transfer intervention, delivered on a mobile platform, on women's empowerment in rural Bangladesh. We recognise that women's empowerment is an essential goal in global public health and the key to sustainable development. It is therefore crucial that, in complex and large-scale nutrition interventions such as the SCC Trial, we measure pathways of impact across the domains of women's empowerment. The 2013 Lancet Series on Maternal and Child Nutrition proposed using interventions that raise women's overall level of empowerment.[10] Research has shown evidence of the impact of various development interventions on women's empowerment.[42–44] A 2018 analysis of surveys from 54 countries observed that in critical aspects of family relationships, four out of five women did not hold agency.[45] Cunningham et al[46] note that women lack not only the resources to make significant decisions leading to better health and nutritional outcomes but also, when disempowered, lack the autonomy and decision-making power within the household to make these critical decisions, having a multiplier effect on family and community.

This study has several strengths; first, it is embedded within a cluster RCT that aims to assess the effectiveness of a nutrition BCC combined with unconditional cash transfers in reducing the prevalence of stunting. Another strength lies in the use of a mixed-methods approach—obtaining qualitative and quantitative data. We gain further advantage from the use of tools tailored specifically for this intervention; tools that are based on formative research, wide-ranging literature review and piloted in Bangladesh, providing accurate, comprehensive and appropriate methodology.

The Pro-WEAI is designed to assess the impact evaluation of agricultural development projects.[47] The SCC Trial is not an agricultural intervention; however, women in Bangladesh characteristically do postharvest activities and processing yet do not classify themselves as agricultural workers. It is in this context that we are using the index to survey women in Bangladesh whose livelihoods are bound to the agricultural sector.[48] Alkire[49] describes the Pro-WEAI as an 'information platform' that is applicable in broader contexts other than exclusively agricultural interventions.[49] The core elements of the SCC women's empowerment index, a customised tool, will provide translatable results—using measures and indices that allow global comparisons.

When the Pro-WEAI is used in full—women and men are interviewed—this enables the calculation of the Gender Parity Index, which measures empowerment and disempowerment within the household. The limitation of not interviewing men is that we may not capture the sense of intrahousehold inequality—are the women at least as empowered as the men in their households? We will account for this by ensuring that qualitative exploration delves into the extent to which the disempowerment observed is attributable to gender norms versus other causes. The SCC Trial does not directly focus on women's empowerment nor contest existing gender norms or patriarchal power structures. However, the study participants—the recipients of the cash transfer, nutrition counselling and smartphone—are women. This study will identify where gaps in empowerment occur and will highlight key areas of disempowerment and which domains help facilitate or hinder nutrition outcomes.

**Author affiliations**
[1]Sydney School of Public Health, Faculty of Medicine and Health, The University of Sydney, Sydney, New South Wales, Australia
[2]Division of Nutritional Sciences, Cornell University, Ithaca, New York, USA
[3]Centre for Health Economics Research and Evaluation, University of Technology Sydney, Sydney, New South Wales, Australia
[4]Maternal and Child Health Division, International Centre for Diarrhoeal Disease Research Bangladesh, Dhaka, Bangladesh
[5]International Centre for Diarrhoeal Disease Research Bangladesh, Dhaka, Bangladesh
[6]The Sax Institute, Sydney, New South Wales, Australia
[7]Nutrition and Clinical Services Division, International Centre for Diarrhoeal Disease Research Bangladesh, Dhaka, Bangladesh
[8]School of Science and Health, Western Sydney University, Penrith, New South Wales, Australia

**Acknowledgements** We gratefully acknowledge the team at the International Food Policy Research Institute for their assistance and for sharing the Project-Level Women's Empowerment in Agriculture Index. I also thank the seminar attendees at the Sydney School of Public Health's Global Health and Nutrition Research Collaboration for providing valuable feedback to my presentation on the tools and methodology used in this paper.

**Contributors** EKK drafted the paper and designed the study tools and main conceptual ideas. AA and MJD made continuous contributions and supervised the design of the study and the overall writing. All other authors (JFH, TH, TLL, TT, MMH, AI, JK, NBA, SU, NG, SM, MMI, GA, KEA and SEA) were involved in the development of the main study design and methods; read, critically revised and approved the

final manuscript; and met the International Committee of Medical Journal Editors criteria for authorship.

**Funding** The Shonjibon Cash and Counselling Trial is funded by the National Health and Medical Research Council of Australia (GNT 1120507). EKK received seed funding for a field visit from the Faculty of Medicine and Health, University of Sydney. The funders do not have any role in the study design, data collection and interpretation of data.

**Map disclaimer** The depiction of boundaries on the map(s) in this article does not imply the expression of any opinion whatsoever on the part of BMJ (or any member of its group) concerning the legal status of any country, territory, jurisdiction or area or of its authorities. The map(s) are provided without any warranty of any kind, either express or implied.

**Competing interests** None declared.

**Patient consent for publication** Not required.

**Provenance and peer review** Not commissioned; externally peer reviewed.

**ORCID iDs**
Elizabeth K Kirkwood http://orcid.org/0000-0001-8603-4903
Michael John Dibley http://orcid.org/0000-0002-1554-5180
M Munirul Islam http://orcid.org/0000-0002-8780-8760

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
