## [Reviewer comments · BMJ Open]

ARTICLE DETAILS

TITLE (PROVISIONAL)	Assessing the impact of a combined nutrition counselling and cash transfer intervention on women's empowerment in rural Bangladesh: A randomised control trial protocol.
AUTHORS	Kirkwood, Elizabeth; Dibley, Michael; Hoddinott, John Frederick; Huda, Tanvir; Laba, Tracey; Tahsina, Tazeen; Hasan, Mohammad; Iqbal, Afrin; Khan, Jasmin; Ali, Nazia; Ullah, Saad; Goodwin, Nicholas; Muthayya, Sumithra; Islam, M; Ara, Gulshan; Agho, Kingsley; Arifeen, Shams E.; Alam, Ashraful

VERSION 1 – REVIEW

REVIEWER	Reynolds, Sarah University of California Berkeley
REVIEW RETURNED	02-Oct-2020

GENERAL COMMENTS	This study proposes to integrate quantitative and qualitative evidence on if a nutrition mhealth intervention benefits women's empowerment. The conceptualization of empowerment is comprehensive and a model of different pathways by which empowerment occurs is also considered, including short-term and long-term pathways. However, in spite of this very nice high-level conceptualization, the implementation is wanting. Connection to the theoretical framework is lacking, particularly in how the quantitative and qualitative aspects complement each other in testing the theory. Most of the quantitative portion of the research is very detailed, but the qualitative research lacks clarity. Main points In some places (e.g. the abstract introduction), it is not clear if the outcome of importance is nutrition or empowerment. This directional confusion is also in the introduction as well, particularly the paragraph starting in line 107. I think it is ok to leave out of the introduction that women's empowerment can also improve nutrition in a positive feedback loop; this probably is better just in the discussion, or perhaps a sentence regarding the conceptual model. I recommend this because the study question seems to be focusing on if the intervention can improve empowerment, not if empowerment subsequently improves nutrition. There is a lack of clarity around the mobile phone as a part of the intervention or not. The introduction says "Several studies have explored how mobile phone programs influence and enhance communication between women and men." (Line 120) but your control group also gets mobile phones, so this does not seem not relevant to your experiment. "When a woman is the target of a program and recipient of the mobile phone, this can challenge
---

gender norms within relationships and exacerbate gender disparities” is similarly problematic (line 125). Perhaps something like: When a woman receives additional resources (referring to the cash transfer and not the phone) and is the target of a program that emphasizes women’s domestic role, these can exacerbate gender disparities.” In figure 1, the mobile phone does not seem like it should be in the intervention column since the control group gets the mobile phone. You deal with this better in the description about the Bangladesh context, where you describe mHealth as the intervention rather than the phone itself.

Most importantly, it is not clear how the quantitative and qualitative studies complement each other. The quantitative analysis is very straightforward, a basic comparison without additional exploration of mechanisms other than different outcomes, though it is nice that there are so many outcomes covering many different facets of empowerment. There are many questions that could be asked from a quantitative perspective, but these questions are not indicated in the research plan. What if women do not recommend another individual in the household to be interviewed? What are the specific questions beyond: “women and men’s perceived and experienced change in empowerment” and “explore the myriad of ways that women and men perceive and describe empowerment.” What hypotheses are you looking to test/confirm in your qualitative data? These should be differentiated from the quantitative hypotheses. If you are looking to test the same hypotheses through triangulation, what will be your conclusion if you find contradictory results in the qualitative and quantitative methods?

Additional comments:

Qualitative research should have enough information for a COREQ checklist or something similar.
<https://link.springer.com/article/10.1007/BF02820685> It is not clear how participants will be recruited or even a target sample size.

This is confusing: The short term outcome is woman’s acquisition of new BCC/nutrition knowledge, which in turn as a long-term outcome, contributes to increased input into and decision making power about health and nutrition choices for herself and her family (Indicator 4). (line 187) It seems you just have 1 timeline – 24 months. Is there a short-term measurement and a long-term measurement? If the long-term outcome refers to the feedback loop, see previous section above.

Figure 1 is much more complex than what you are able to test in the intervention. Maybe highlight which parts are covered by the quantitative and qualitative analysis.

	How do you know gestational age is <90 days? Ultrasound? What sort of bias is introduced by excluding women who learn they are pregnant after 90 days? Line 242 – are there 3 trial arms or are these three interventions all given to the treatment group? Not clear. It is not clear why the primary and secondary hypotheses are divided as such. In particular, a decrease in IPV seems very important and could be elevated to a primary outcome. I would think your primary and secondary outcomes should be linked to their importance in the theoretical model or some sort of data reliability issue, but I don't see those connections. Outcome measures: are you only using 1 composite measure and not examining the sub-measures? It would be useful to see the questionnaire of outcome variables. Decision-making ability about health and nutrition choices seems more like an outcome related to the effectiveness of the main intervention rather than an exploration of empowerment. The rest of your paper focuses on intrahousehold type empowerment between spouses rather than empowerment in the world. I suggest cutting this to keep focused on intra-household dynamics. Your outcome variables are very broad in covering all kinds of empowerment measures. I would like one paragraph/section/sentence – whichever makes the most sense -- on each of the 6 aspects of the Pro-WEAI. Keep them in the same order throughout the paper. The IPV section in the introduction is not well-integrated and seems rather disconnected from the other outcomes discussed. You could add heterogeneity analysis to the quantitative analysis: do women who are more or less empowered at baseline benefit more from the intervention? Are women with mothers-in-law in the household more or less empowered? I hope these suggestions are useful – best wishes for a successful implementation of this important research!
--	---

REVIEWER	Kulkarni, Bharati National Institute of Nutrition
REVIEW RETURNED	25-Dec-2020

GENERAL COMMENTS

The trial is embedded in another cluster randomized trial aimed at assessing the impact of a combined nutrition counselling and cash transfer intervention on stunting in rural Bangladesh. The current protocol describes the assessment of an additional outcome 'women empowerment' within the framework of the larger SCC trial. It is surprising that the authors have not referenced the main trial protocol paper (<https://doi.org/10.1186/s12889-020-09780-5>) although they have mentioned that a separate protocol paper provides details of the design. This is probably because the current manuscript was submitted before the main SCC trial protocol manuscript. But it would have been prudent to submit this manuscript referencing the main SCC trial protocol paper where the details about the trial interventions, sample size estimation etc are described. Understandably, there is a significant overlap in the two papers as both describe the same study except an additional component of the theory of change about the outcome of women's empowerment. I do not think this protocol of an additional outcome merits a separate manuscript as it does not make a significant contribution to literature in the current format.

Many aspects of the study design are unclear in the current manuscript. For example:

1) What is the basis for the primary outcome – average 20 percent increase in empowerment score? Is it meaningful to impact any welfare related measures? What is the power of the study to detect this difference with a given sample size?

2) The authors intend to assess the impact of the intervention on empowerment using WEAI which has been designed to assess the empowerment in agriculture. As the intervention includes cash transfer with nutrition counseling, they should also assess more specifically women's empowerment in nutrition (Please see: Narayanan, Sudha, et al. "Developing the women's empowerment in nutrition index in two states of India." *Food Policy* 89 (2019): 101780.)

3) The Introduction should clearly state that the current study is a part of the SCC Trial.

4) The statement 205 describing IPV is unclear.

5) Study outcomes lines 250-253: The outcomes should not include direction of effect. For example, the outcome is 'control over income and economic resources', not 'an increase in control over income and economic resources'.

6) Social desirability questions (Appendix table 3): it is not clear whether this will be scored and what score would qualify as social desirability.

7) It is mentioned that a project specific module Pro-WEAI would be used. Has this been validated?

8) Line 373-4. More details about the DSMB need to be provided including the composition, any interim analyses planned. What are the adverse events that are anticipated? How will these be recorded / reported?

9) Details on the planned statistical analyses are not provided.

VERSION 1 – AUTHOR RESPONSE

Reviewer 1

General Comment: Most of the quantitative portion of the research is very detailed, but the qualitative research lacks clarity.

Response: We thank the reviewer for the suggestion. We hope that with edits, as detailed below, that the paper now clearly shows the way the qualitative and quantitative aspects complement each other, and the more detailed qualitative research section is clear and concise.

Comment: In some places (e.g. the abstract introduction), it is not clear if the outcome of importance is nutrition or empowerment.

Response: We agree to this comment and have adjusted the text in the abstract as follows: "Introduction: There is growing interest in assessing the impact of health interventions, particularly when women are the focus of the intervention, on women's empowerment. Globally, research has shown that interventions targeting nutrition, health, and economic development can affect women's empowerment. Evidence suggests that women's empowerment is also an underlying determinant of nutrition outcomes. Depending on the focus of the intervention, different domains of women's empowerment will be influenced for example an increase in nutritional knowledge, or greater control over income and access to resources." (Line 51) The introduction is acknowledging the intrinsic connection and positive feedback loop that women's empowerment and nutritional status share.

Comment: This directional confusion is also in ... the paragraph starting in line 107.

Response: We thank the author for this comment. We have removed the paragraph from the introduction and have moved some relevant sentences to the discussion.

Comment: There is a lack of clarity around the mobile phone as a part of the intervention or not.

Response: We thank the reviewer for this comment and adjusted the text to clarify the intervention: "When a woman receives additional resources, such as cash transfers and is the target of a mHealth program this can challenge gender norms within relationships and exacerbate gender disparities" (Line 117)

Comment: In Figure 1, the mobile phone does not seem like it should be in the intervention column since the control group gets the mobile phone.

Response: Thank you, this is a valid point as both the intervention and control group receive a mobile phone. Both groups receive a mobile phone to minimize the non-specific effects of owning a mobile communication device. We have however adjusted the text to read as below (removing the emphasis on the mobile phone itself): "Women receive an mHealth interactive app – messages, audio and video, quizzes – as well as counselling from the call centre" (Line 172). We have adjusted Figure 1 accordingly.

Comment: It is not clear how the quantitative and qualitative studies complement each other... There are many questions that could be asked from a quantitative perspective, but these questions are not indicated in the research plan.

Response: We assume the reviewer was referring to a qualitative perspective and have answered accordingly. Our study's qualitative data will complement and contextualize the quantitative findings, for example men and other family members are not interviewed for the quantitative survey. We will explore women's, men's and other household members lived experience and perceived changes relating to the intervention, which cannot be captured through the quantitative data. We have attached the qualitative interview guidelines as a supplementary file.

Comment: What if women do not recommend another individual in the household to be interviewed?

Response: Based on our experience in similar studies in Bangladesh, we anticipate that a significant proportion of women will recommend someone in their household to be interviewed. If

women do not recommend other household members to be interviewed, we will still have a sufficient sample for our qualitative data collection.

Comment: What are the specific questions beyond: “women and men’s perceived and experienced change in empowerment” and “explore the myriad of ways that women and men perceive and describe empowerment.”

Response: We have added the qualitative interview guidelines as supplementary material. The guideline has questions such as “How has this nutrition knowledge influenced your decision making related to healthcare for your household?”, “Are you able to use the money for whatever you wish to?”, “Have you noticed any changes in the way that you and your husband communicate since this project has started?”.

Comment: What hypotheses are you looking to test/confirm in your qualitative data? ... If you are looking to test the same hypotheses through triangulation, what will be your conclusion if you find contradictory results in the qualitative and quantitative methods?

Response: There is no separate hypothesis for the qualitative data. The hypothesis for this protocol is that in a community-based cluster randomised control trial of a mobile phone-based nutrition behaviour change communication (BCC) and unconditional cash transfer, given to women in low-income families in rural Bangladesh, women’s empowerment will increase as measured by an increase in mean women’s empowerment scores (by an average percentage of 20) from the baseline to the end of the 24-month intervention, compared to women in the control arm (Line 157). Our qualitative data will complement the quantitative and examine the same hypothesis.

Comment: Qualitative research should have enough information for a COREQ checklist or something similar.

Response: We are reporting on a mixed methods study protocol in this paper and have followed the SPIRIT and Tidier guidelines, the international standard for the reporting clinical trial protocols, as suggested by the journal (Line 217).

Comment: It is not clear how participants will be recruited or even a target sample size.

Response: We will select our participants for qualitative survey based on purposive sampling technique from within the households participating in the survey. Participants include women and family members such as husbands and mothers-in-law.

Comment: Is there a short-term measurement and a long-term measurement? If the long-term outcome refers to the feedback loop, see previous section above.

Response: We thank the reviewer for this question and opportunity to clarify Figure 1. We have adjusted the box headers in Figure 1; we have changed “short term outcomes” to “intermediate outcomes” and “long term outcomes” to “outcomes” as we are not using short term measurements. We have also adjusted any related text accordingly.

Comment: Figure 1 is much more complex than what you are able to test in the intervention. Maybe highlight which parts are covered by the quantitative and qualitative analysis.

Response: Figure 1 explains the theory of change and as mentioned previously, the qualitative data will complement the quantitative data. There is no distinct difference in what we are testing with the intervention so we cannot separate the qualitative and quantitative.

Comment: How do you know gestational age is <90 days? Ultrasound?

Response: We are following a standard protocol, based on our experience in similar RCTs in Bangladesh. We are not using ultrasound to determine gestational age.

We will conduct household surveillance and list all the women of reproductive age currently not pregnant. Our surveillance worker will then conduct door-to-door bi-monthly visits to identify women missing two menstrual periods in a row. All such woman will undergo a pregnancy test with a sensitive pregnancy urine test kit (Excel®). We will invite women to participate who test positive in the study. We will only enroll women who have become pregnant since our last visit (<60 days).

Comment: What sort of bias is introduced by excluding women who learn they are pregnant after 90 days?

Response: We do not expect selection bias from only recruiting women who are in their first trimester of pregnancy due to the active pregnancy surveillance methods detailed above. If the recruitment

used a passive system and participants self-detected their pregnancy it is possible that less empowered women might present later. But our active home-based surveillance system will provide equal access to pregnancy tests irrespective of the women's autonomy to seek health care services. Comment: Line 242 – are there 3 trial arms or are these three interventions all given to the treatment group? Not clear.

Response: Thanks to the reviewer for this question. We have changed the text to clarify as follows: "The SCC Trial intervention arm receives: 1) nutrition BCC delivered on a specially tailored app on a smartphone (audio, video, and animation), 2) direct nutrition counselling from a call centre and 3) unconditional cash transfer of 1000 Taka (USD 12.50) received monthly via BKash mobile banking app." (Line 238)

Comment: It is not clear why the primary and secondary hypotheses are divided as such. In particular, a decrease in IPV seems very important and could be elevated to a primary outcome. I would think your primary and secondary outcomes should be linked to their importance in the theoretical model or some sort of data reliability issue.

Response: Our primary hypothesis is to measure the impact of the SCC Trial on women's empowerment. Our secondary hypothesis reflects the individual components that combine to form our composite index. Changes in IPV and whilst not the main outcome, is a very important outcome and will be explored quantitatively and qualitatively in great detail. The primary and secondary outcome are linked to the theoretical model of empowerment and the theory of change. The intervention will impact individual indicators in different ways and therefore the secondary outcomes are the individual indicators that form the composite index or primary outcome measure.

Comment: Outcome measures: are you only using 1 composite measure and not examining the sub-measures? It would be useful to see the questionnaire of outcome variables.

Response: We thank the reviewer for this question. Our main outcome measure is a change in empowerment as measured by the composite measure. This composite measure is made up of 6 individual indicators or sub-measures which will also be examined individually and complemented with the qualitative research. Please find the questionnaires for each of the indicators as a supplementary file.

Comment: Decision-making ability about health and nutrition choices seems more like an outcome related to the effectiveness of the main intervention rather than an exploration of empowerment. The rest of your paper focuses on intrahousehold type empowerment between spouses rather than empowerment in the world. I suggest cutting this to keep focused on intra-household dynamics.

Response: Whilst decision-making ability about health and nutrition choices may seem more like an outcome related to the main intervention rather than an exploration of empowerment, the focus of our questions is on women's agency and changes in agency over the course of the intervention.

Our health and nutrition questions focus on a woman's instrumental agency, and their ability to have input into the health care, nutrition related decision-making process. The indicator (Indicator 4 - Decision-making power on nutrition and health care) is also linked to intrahousehold relationships which we explore under Indicator 5 - Respect among household members, and both will be complemented with our qualitative exploration.

Comment: I would like one paragraph/section/sentence – whichever makes the most sense -- on each of the 6 aspects of the Pro-WEAI. Keep them in the same order throughout the paper.

Response: We thank the reviewer for this comment. Due to word limitations some of the indicators were combined, we have now separated the indicators for clarity and have added one sentence on each aspect of the Pro-WEAI (Line 259) and we will keep them in order throughout the paper.

Comment: The IPV section in the introduction is not well-integrated and seems rather disconnected from the other outcomes discussed.

Response: Thank you for this comment. IPV is not the sole focus of our assessment of the impact of SCC intervention on women's empowerment and due to word limitations, we have had to be concise. We have added text to the introduction as follows "Evidence suggests that the combination of cash transfers and behaviour change communication (BCC), can increase women's bargaining power and poverty-related emotional well-being and lead to a reduction in IPV" (Line 109).

Comment: You could add heterogeneity analysis to the quantitative analysis: do women who are more or less empowered at baseline benefit more from the intervention? Are women with mothers-in-law in the household more or less empowered?

Response: We agree with the reviewer that we should examine key baseline factors as potential modifiers of the women's empowerment response to the intervention. We have expanded the description of the analysis of the quantitative data as follows:

"Quantitative Data Analysis

Data analysis will be by intention to treat. The women's empowerment composite index scores will be categorized as the percentage of women empowered or disempowered. For each empowerment indicator individuals are classified as adequate or inadequate based on Pro-WEAI predetermined thresholds. Women's are considered empowered if 4 out of 6 indicators are adequate. We will also analyze the impact of SCC intervention on individual indicators to assess increases or decreases in empowerment scores, as not all indicators will respond or be impacted in the same way.

Analyses will be conducted at the mother-infant dyad level but will be adjusted for the cluster randomization (Hayes & Moulton, 2009). Primary analyses will compare the prevalence of women's empowerment at the end of the trial using Pearson's chi-square tests and 95% confidence intervals for the group difference, adjusted for clustering and generalized linear mixed models for non-continuous outcomes (e.g. logistic mixed models for binary outcomes e.g. percentage of women's empowerment). Models will include treatment group as a fixed effect, infants as a random effect to account for repeated measurements, and community-cluster as a random effect to account for cluster effects. We will also assess if the women's empowerment level at baseline, age, education and presence or absence of mother-in-law in the household modify the empowerment response to the intervention by testing for interactions between the intervention and these factors. STATA® will be used for all analyses." (Line 381).

Reference

Hayes HJ, Moulton LH. Cluster randomised trials. Boca Rato:US: Taylor & Francis; 2009.

Reviewer 2

Comment: The trial is embedded in another cluster randomized trial aimed at assessing the impact of a combined nutrition counselling and cash transfer intervention on stunting in rural Bangladesh...the authors have not referenced the main trial protocol paper.

Response: We thank the reviewer for this suggestion. The SCC Trial Protocol paper was not published at the time of submission of this paper. The reference to the main trial protocol has now been added (Line 131). The SCC trial will measure the impact of combined nutrition counselling and cash transfers on stunting and will not measure women's empowerment component. We will analyze women's empowerment as a stand-alone study using theoretical framework discussed in this paper. We hope that the revised version of this manuscript will make a significant contribution to the literature on the impact on complex and innovative nutrition interventions on women's empowerment.

Comment: What is the basis for the primary outcome – average 20 percent increase in empowerment score?

Response: Previous reports of studies examine the impact of a single interventions on women's empowerment indicate an impact of 10 to 15% (Quisumbing et al 2020) We have hypothesized a larger impact because we are testing multiple interventions.

Quisumbing, Agnes R. and Ahmed, Akhter U. and Hoddinott, John and Pereira, Audrey and Roy, Shalini, Designing for Empowerment Impact in Agricultural Development Projects: Experimental Evidence From the Agriculture, Nutrition, and Gender Linkages (ANGeL) Project in Bangladesh (August 14, 2020). IFPRI Discussion Paper 1957, 2020, Available at SSRN:

<https://ssrn.com/abstract=3674113>

Is it meaningful to impact any welfare related measures?

Response: We are not sure what the reviewer means by this question and are therefore unable to respond.

What is the power of the study to detect this difference with a given sample size?

Response: The sample size is fixed by the primary hypothesis in the main trial. This means we have a total of 104 cluster with 21 participants per cluster (Huda et al 2020). There are no reports of intra cluster correlation coefficients for the women's empowerment. A recent survey from Bangladesh (Quisumbing et al 2020) suggests about 25% of women are empowered using the same indicators we plan for our trial. We calculated the minimal detectable difference assuming 90% power for a range of possible icc values.

At a low icc of 0.001 we have 90% power to detect a 5% improvement in women's empowerment. The minimum detectable difference decreases as the assumed icc and with a very high icc of 0.2 our fixed sample size could still detect a 22% increase in women's empowerment with 90% power and a 19% increase in women's empowerment with 80% power.

We have added the following paragraph to clarify the adequacy of our sample size.

"Sample size and power"

The sample size for our trial is fixed by the primary hypothesis in the main Shonjibon Cash and Counselling trial, which estimated a total sample of 2184 mother-infant pairs from 104 clusters (Huda et al 2020). We can find no reports of intra cluster correlation coefficients (icc) for the women's empowerment indicators we plan to use. Therefore, we estimate that the fixed trial sample size will provide at least 80% power to detect a 20% increase in women's empowerment assuming 30% of women are empowered (ref) and a high icc of 0.2. Assuming a lower icc of 0.05 we will have 90% power to detect a 13% increase in women's empowerment." (Line10)

Reference

Huda, T.M., Alam, A., Tahsina, T. et al. Shonjibon cash and counselling: a community-based cluster randomised controlled trial to measure the effectiveness of unconditional cash transfers and mobile behaviour change communications to reduce child undernutrition in rural Bangladesh. *BMC Public Health* 20, 1776 (2020). <https://doi.org/10.1186/s12889-020-09780-5>

Quisumbing, Agnes R. and Ahmed, Akhter U. and Hoddinott, John and Pereira, Audrey and Roy, Shalini, Designing for Empowerment Impact in Agricultural Development Projects: Experimental Evidence From the Agriculture, Nutrition, and Gender Linkages (ANGeL) Project in Bangladesh (August 14, 2020). IFPRI Discussion Paper 1957, 2020, Available at SSRN: <https://ssrn.com/abstract=3674113>

Comment: The authors intend to assess the impact of the intervention on empowerment using WEAI which has been designed to assess the empowerment in agriculture. As the intervention includes cash transfer with nutrition counseling, they should also assess more specifically women's empowerment in nutrition (Please see: Narayanan, Sudha, et al. "Developing the women's empowerment in nutrition index in two states of India." *Food Policy* 89 (2019): 101780.)

Response: We thank the author for this suggestion. The WEAI is designed to assess the impact of interventions on empowerment in agriculture. The majority of our study participants work in agriculture and the women participate in post-harvest activities and their livelihoods are clearly tied to the agricultural sector. The Project Specific WEAI is an updated version of the WEAI that measures women's empowerment in project specific contexts, and has additional module designed to assess women's empowerment and agency in the domains of health and nutrition decisions. The health and nutrition module is comprised of seven indicators that assess a woman's agency and decision-making ability and covers key components that we will monitor. We have carefully considered the WENI as suggested by the reviewer and whilst the WENI and Pro-WEAI have common ground, for our purposes we feel the Pro-WEAI health and nutrition module covers questions in alignment with the focus of our study and fits with the rest of the Pro-WEAI modules we are using.

Comment: The Introduction should clearly state that the current study is a part of the SCC Trial.

Response: We have amended the manuscript as follows: "We plan to conduct a study, embedded in a cluster randomised control trial that assesses a multifaceted nutrition intervention on childhood stunting - Shonjibon Cash and Counselling Trial (SCC). This protocol paper presents the way we will measure the impact of the SCC trial women's empowerment. (LINE 131)

Comment: The statement 205 describing IPV is unclear.

Response: We have amended the text as follows, “IPV includes behaviour that is physical, psychological, sexual or abusive or controlling in nature” (Line 202)

Comment: Study outcomes lines 250-253: The outcomes should not include direction of effect.

Response: We have modified the study outcomes as follows: “Secondary study outcomes will include 1) control over income and economic resources; 2) input and decision-making power in nutrition and health care choices; 3) experience and attitude toward intimate partner violence.” (Line 247)

Comment: Social desirability questions (Appendix table 3): it is not clear whether this will be scored and what score would qualify as social desirability.

Response: We will use a score and we have added the following text in the sub-section on Data Collection Methods – Quantitative data

“We will use the validated short version of the Marlowe-Crowne Social Desirability Scale [ref], which is based on a subset of 13 items from the original scale. We will calculate a social desirability score by adding up the number of socially desirable answers, out of the 13 questions. The potential range of the score will be from 0-13 and we will create three categories with a score of 0-4 graded as a low score, 5-9 as a medium score, and 10-14 as a high score.” (Line 327)

Reynolds WM. Development of reliable and valid short forms of the Marlowe-Crowne Social Desirability Scale. J Clin Psychol 1982;38(1):119-125. [doi: 10.1002/1097-4679(198201)38:1<119::AID-JCLP2270380118>3.0.CO;2-I]

Comment: It is mentioned that a project specific module Pro-WEAI would be used. Has this been validated?

Response: The Pro-WEAI has been validated, this paper describes the adaptation and validation of a project-level WEAI. Five projects in the portfolio were used to validate this version of the pro-WEAI, with 4 of the projects piloted in Bangladesh.

<https://www.sciencedirect.com/science/article/pii/S0305750X19301706>

Comment: Line 373-4. More details about the DSMB need to be provided including the composition, any interim analyses planned. What are the adverse events that are anticipated? How will these be recorded / reported?

Response: We have provided further information re the DSMB as suggested: “An independent data and safety monitoring board (DSMB) will be formed to assess the completeness and quality of data and to ensure data is compliant with recruitment and retention goals. The DSMB will also assess any factors that might affect the study outcome or compromise the confidentiality of the trial data. Any unintended effects of the trial will be reported to the board.” (Line 408)

Potential adverse events that could occur, from a gender-based perspective, could be an increase in relationship conflict, however, based on our pilot study (<https://mhealth.jmir.org/2018/7/e156/>), we find this to be unlikely and studies have shown an improvement in spousal relationships.

Comment: Details on the planned statistical analyses are not provided.

Response: We thank the reviewer for this comment and have added a section on planned statistical analysis (Line 381)

VERSION 2 – REVIEW

REVIEWER	Reynolds, Sarah University of California Berkeley
REVIEW RETURNED	24-Feb-2021
GENERAL COMMENTS	Thank you for your revisions. I find my concerns have been thoughtfully addressed. Best wishes for successful implementation.
REVIEWER	Kulkarni, Bharati National Institute of Nutrition

REVIEW RETURNED	13-Mar-2021
GENERAL COMMENTS	I wish to thank the authors for accepting my suggestions and making appropriate changes in the manuscripts. My concerns have been adequately addressed in the current version and I have only a few very minor comments.  1. In strengths and limitations of the study, third bullet point " Designed specifically ... global comparison" is unclear. Please rephrase for better clarity. 2. Line 406: 'Women's are considered...' is grammatically incorrect. 3. Patient and public involvement: Use of word 'patient' in this section is not appropriate, please use 'participants'. 4. Lines 448-450: It is not clear whether the dataset would be shared with other researchers e.g. shared in a repository. Please clarify.

VERSION 2 – AUTHOR RESPONSE

I have uploaded the clean and marked copy of the manuscript and addressed reviewer 2's comments as follows:

1. In strengths and limitations of the study, third bullet point " Designed specifically ... global comparison" is unclear. Please rephrase for better clarity.

I have amended to:

"We have designed specifically tailored tools, based on a theory of change and utilising an internationally validated index, that has been piloted in Bangladesh.

2. Line 406: 'Women's are considered...' is grammatically incorrect.
Corrected to read "Women are considered..."

3. Patient and public involvement: Use of word 'patient' in this section is not appropriate, please use 'participants'.

Have amended to use the word participants and not patient

4. Lines 448-450: It is not clear whether the dataset would be shared with other researchers e.g. shared in a repository. Please clarify.

I think the reviewer is referring to Line 427 - I have added:

"The datasets generated and analysed during the study will be available from the corresponding author on reasonable request."